# Autofluorescence Imaging of Living Yeast Cells with Deep-Ultraviolet Surface Plasmon Resonance

Che Nur Hamizah Che Lah [1], Hirofumi Morisawa [2], Keita Kobayashi [3], Atsushi Ono [1,3,4,*], Wataru Inami [1,3,4] and Yoshimasa Kawata [1,2,3,4]

1 Graduate School of Science and Technology, Shizuoka University, 3-5-1 Johoku, Naka-ku, Hamamatsu 432-8561, Japan; che.nur.hamizah.18@shizuoka.ac.jp (C.N.H.C.L.); inami.wataru@shizuoka.ac.jp (W.I.); kawata@eng.shizuoka.ac.jp (Y.K.)
2 Department of Mechanical Engineering, Shizuoka University, 3-5-1 Johoku, Naka-ku, Hamamatsu 432-8011, Japan; morisawa.hirofumi.14@shizuoka.ac.jp
3 Graduate School of Integrated Science and Technology, Shizuoka University, 3-5-1 Johoku, Naka-ku, Hamamatsu 432-8561, Japan; kobayashi.keita.17@shizuoka.ac.jp
4 Research Institute of Electronics, Shizuoka University, 3-5-1 Johoku, Naka-ku, Hamamatsu 432-8561, Japan
* Correspondence: ono.atsushi@shizuoka.ac.jp

**Abstract:** Autofluorescence in living cells on aluminum thin film was excited with deep-ultraviolet surface plasmon resonance (deep-UV SPR). Deep-UV SPR under aqueous medium was excited with Kretschmann configuration by using a sapphire prism. Deep-UV SPR is one of the promising techniques for high-sensitive autofluorescence imaging of living cells without staining. Label-free observation is significant for the structural analysis of living cells. We demonstrated the high-sensitive autofluorescence imaging of living yeast cells with deep-UV SPR. We applied a high refractive index prism, such as sapphire, which is suitable for the observation of specimens in aqueous medium, to excite deep-UV SPR. Although typical autofluorescence from living cells is buried in background noise, deep-UV SPR enhances the autofluorescence signal. The deep-UV SPR excitation of an aluminum thin film through a sapphire prism was investigated theoretically and experimentally. It showed that the fluorescence intensities are increased 2.8-fold. Deep-UV SPR enhanced the autofluorescence of cell structures, and yeast cells were found to be very sensitive. As a result, for water-immersed specimens, the sapphire-prism-based Kretschmann configuration excited SPR in deep-UV. Findings from this study suggest that deep-UV SPR can be considered an effective technique for attaining high-sensitivity observation of biological samples.

**Keywords:** autofluorescence; surface plasmon resonance; deep-ultraviolet; bio-imaging; yeast; Kretschmann configuration

## 1. Introduction

There has been a rapid advancement in bio-imaging with the development of fluorescent technologies, thereby contributing to the analysis of molecular interactions [1] and cellular functions [2]. Bio-imaging is significant for studying cellular processes [3] and visualizing living cells [4,5]. Recent developments in bio-imaging include surface plasmon resonance (SPR)-based high-sensitive imaging, autofluorescence microscopy [6–8], near-infrared reduced illuminance autofluorescence imaging (NIR-RAFI) [9], Raman microscopy [10], surface-enhanced Raman scattering (SERS) [11,12] and coherent anti-Stokes Raman scattering (CARS) microscopy [13–15] as label-free imaging techniques. Over the last few years, a possible link between biological molecules and bio-imaging has led to the development of an increasing number of biosensors, as they can be used to monitor living cells without staining [16].

Significant attention has been focused on the exploitation of SPR in the deep-ultraviolet (deep-UV) spectral region, which is particularly important in the analysis of organic and

biomolecules [17]. For example, surface plasmon applications that focus on the photon energy of deep-UV light include biosensors [18–20], increased photoemission [21,22], fluorescence enhancement [17–19,23–25], surface-enhanced Raman scattering [11,12], tip-enhanced Raman scattering [26] and surface-enhanced resonance Raman scattering (SERRS) [27].

One of the features of deep-UV light is that the autofluorescence of light can be excited in most kinds of biological structures in cells [24,25]. Autofluorescence is an intrinsic phenomenon, whereby light is emitted from organisms. Thus, it is possible to identify the material with fluorescence spectroscopy in the deep-UV region. Our group demonstrated the observation of fixed cells MC3T3-E1 with autofluorescence enhancement with deep-UV SPR [16].

In this study, we are focusing on highly sensitive autofluorescence imaging of living cells through deep-UV SPR excitation. We demonstrated the applicability of deep-UV SPR to the fluorescence imaging of living yeast cells without staining. We believe that the enhanced autofluorescence imaging in the Kretschmann configuration is superior in avoiding substantial damage in deep-UV irradiation.

## 2. Theoretical Analysis

In the present study, aluminum was selected as a metal film because its dielectric constant has a negative real part and a small imaginary part in the deep-UV wavelength. The SPR of Al thin films evaporated on sapphire prisms was studied in this research.

Figure 1 shows a schematic diagram of autofluorescence imaging of living cells cultured on the aluminum thin film. The deep-UV light was illuminated on aluminum thin film through a sapphire prism in the Kretschmann configuration. Autofluorescence image of living cells was observed through the objective lens. All refractive indices used in this configuration were listed in Table 1.

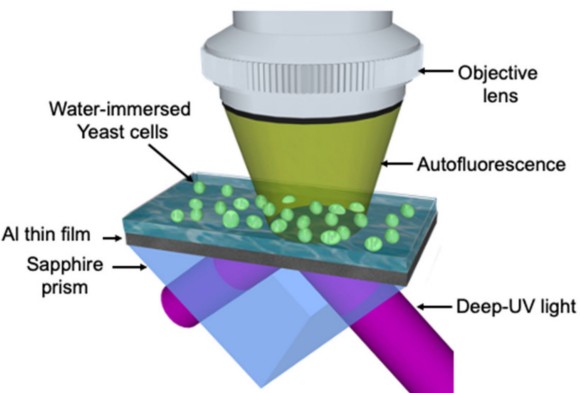

**Figure 1.** Schematic diagram of living yeast cell observation using deep-UV SPR.

**Table 1.** Refractive indices at 266 nm wavelength used in the Kretschmann configuration.

| Material | Refractive Index@266nm |
|---|---|
| Aluminum | 0.21 + i3.14 |
| Alumina | 1.83 |
| Quartz prism | 1.52 |
| Sapphire prism | 1.83 |
| Water | 1.37 |

Deep-UV light has a sufficient energy to excite autofluorescence of biological molecules. The typical autofluorescence is a weak signal; however, surface plasmon enhances the electric field localized on the aluminum surface, and it only illuminates the cell surface attached to the aluminum. Since an 18-fold fluorescence enhancement was demonstrated by deep-UV SPR [17], higher-contrast image will be obtained with the autofluorescence

enhancement of organelles in cells. This is expected to result in high-sensitive imaging without staining the cells, and it is useful for analyzing cell division and growth cycles.

In general, a quartz prism is used for deep-UV SPR in the Kretschmann configuration [16,17,24,25]. However, quartz's refractive index is insufficient to excite surface plasmons in water-immersed living cells at deep-UV wavelengths. As a result, in this paper, we propose using a sapphire prism with a higher refractive index in a Kretschmann configuration. Comparisons between quartz and sapphire prisms were performed using the theoretical incidence angle dependence of reflectance on an aluminum thin film to investigate the deep-UV SPR conditions at the wavelength of 266 nm. The optimal aluminum thickness is 21 nm [24]. Since aluminum surface can be easily oxidized and covered with a thickness of a few nanometers' alumina, we included the alumina film with 6 nm thickness in the calculation. Assuming a Kretschmann configuration, the reflectance dependence on the incident angle and the refractive index of specimens were calculated with Fresnel equations with multi-layered structures [28]. We used all refractive indices of the quartz prism ($SiO_2$), sapphire prism ($Al_2O_3$), aluminum (Al) thin film and oxidized layer of alumina ($Al_2O_3$) at 1.52, 1.83, 0.21 + i3.14 and 1.83, respectively. The refractive index of the top layer on alumina was varied from 1.0 to 1.8 for the consideration of various kinds of specimens. The incident light was *p*-polarized at the wavelength of 266 nm.

Figure 2a,b shows the theoretical reflectance dependence on the incident angle and the refractive index of specimen for aluminum thin film on quartz and sapphire prisms, respectively. The reflectance dip indicates the deep-UV SPR excitation conditions. The excitation angle was shifted to a larger angle as the refractive index of the specimen increased. The refractive index range for deep-UV SPR excitation was expanded by applying a sapphire prism. In particular, assuming the observation of water-immersed specimens, the refractive index of water 1.37 at 266 nm wavelength [29], the critical angle of the total reflection shifted over 60 degrees, and no reflectance dip was observed, as shown in the line profile of Figure 2c. This is because the incident wave vector in a quartz prism is not enough to satisfy the matching condition with surface plasmon wave vector on the aluminum surface in aqueous media. Therefore, when in water, deep-UV SPR cannot be excited with a quartz prism. On the other hand, in the case of sapphire prism indicated as Figure 2d, the reflectance dip angle appeared at 62 degrees, and the reflectance value was kept to the minimum. Therefore, the prism of a sapphire is appropriate for deep-UV SPR excitation with water-immersed specimens. It is a great advantage for high-sensitive bio-imaging.

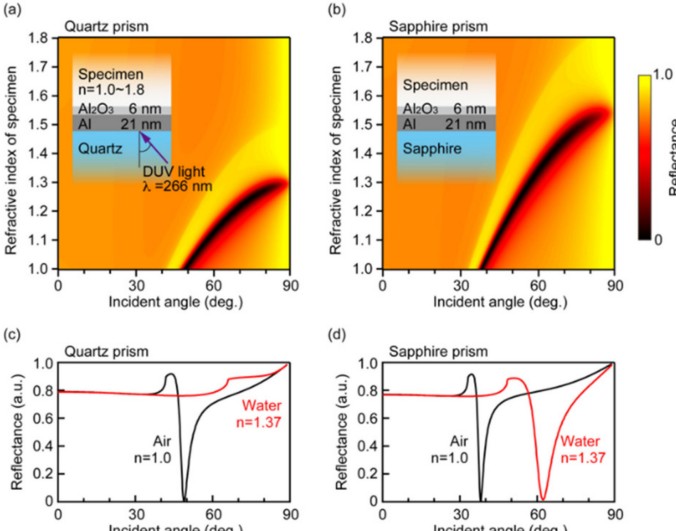

**Figure 2.** The theoretical reflectance dependence on the incident angle and the refractive index of specimen for aluminum thin film on (**a**) quartz and (**b**) sapphire prisms and the line profiles in refractive indices of air and water on (**c**) quartz and (**d**) sapphire prisms.

### 3. Experimental Methods and Results

Subsequent to the theoretical studies, the experimental reflectance of the incident angle dependence for a sapphire prism was measured. Aluminum was directly evaporated on a sapphire prism with 21 nm thickness. It is considered that several nanometers of the evaporated film were oxidized. The deep-UV light source we used was the fourth harmonic of the Nd:YAG laser with a wavelength of 266 nm. The incident polarization was set to *p*-polarization by using a half-wave plate. The reflectance was detected with Si photodiode with a sample rotation. In terms of the results, Figure 3 shows the experimental reflectance dependence on the incident angle for Al thin film with a sapphire prism. The experimental reflectance dips specified by the solid lines were observed at the incident angle of 37 degrees for air and 61 degrees for water. When we assume the natural oxide layer of alumina thickness as 4.5 nm and the remained aluminum thickness as 16.5 nm, the experimental reflectance of incident angle dependence is well matched with the theoretical results indicated in the broken lines.

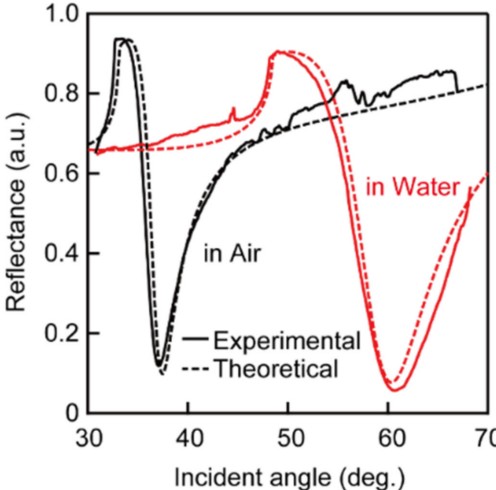

**Figure 3.** Experimental (solid) and theoretical (broken) reflectance dependences on the incident angle in air and in water.

As the first demonstration of water-immersed sample observation using deep-UV SPR with a sapphire prism, we observed fluorescent polystyrene latexes (Fluoresbrite® BB Carboxylate Microspheres, Polyscience, Warrington Township, PA, USA). The suspension of polystyrene latexes, 1 μm in diameter, was dropped on the evaporated aluminum thin film. The deep-UV light incident angle was adjusted to 61 degrees to excite SPR. The fluorescence image was observed through water immersion objective lens (LUMPLFLN60XW, Olympus, Shinjuku City, Japan) of numerical aperture 1.0 with a magnification 60×. Figure 4a,b show fluorescence images of water-immersed polystyrene latexes excited with *p*-polarization and *s*-polarization, respectively. The excitation laser power was 20 μW, and the exposure time was 0.3 s. Figure 4c shows the conventional reflection image of the corresponding area. These images were acquired by ORCA-Flash4.0 V3 digital CMOS camera. In Figure 4a, individual latexes were clearly observed, and the fluorescence intensity was much greater than that without deep-UV SPR shown in Figure 4b. Figure 4d shows the fluorescence intensity line profiles for quantitatively evaluating the enhancement ratio between Figure 4a,b. For the comparison of the enhancement factor, the background signal was subtracted. A comparison of the fluorescence intensities at the position with the peak intensities in Figure 4d confirmed an approximate 2.8-fold increase. In terms of the results, we can conclude that deep-UV SPR enhanced the fluorescence of the water-immersed samples, and it was possible to achieve highly sensitive observation.

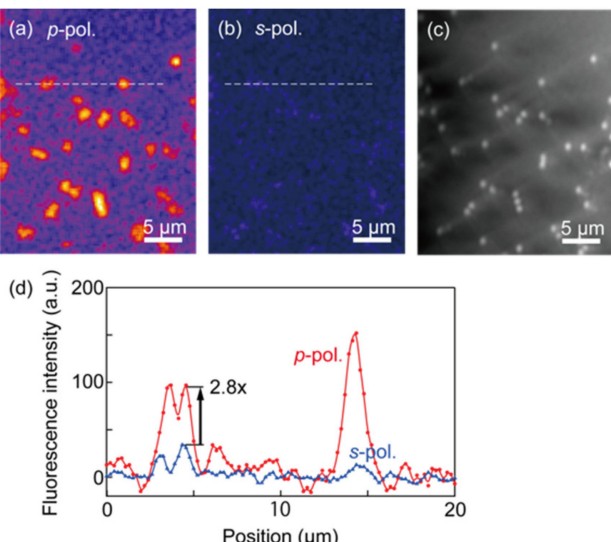

**Figure 4.** The fluorescence images of water-immersed polystyrene latexes excited with (**a**) *p*-polarization, (**b**) *s*-polarization, (**c**) conventional reflection image and (**d**) the fluorescence intensity line profiles between *p*-polarization and *s*-polarization fluorescence images.

Since fluorescence enhancement using a sapphire prism with deep-UV SPR was demonstrated, label-free yeast cells were observed in the same manner. We demonstrated autofluorescence imaging of yeast cells by deep-UV SPR with a quartz prism, but it was observed under the dried cells [16]. We used yeast cells for label-free imaging with deep-UV SPR. Like human cells, yeast cells have a typical structure. Yeast cells are easy to manipulate, grow easily and are suitable for biological research. In this paper, we observed water-immersed living yeast cells using deep-UV SPR with a sapphire prism. This is significant for the practical analysis of cellular functions with bio-imaging.

Figure 5 shows the autofluorescence image of the water-immersed living yeast cells on the aluminum thin film (a) with deep-UV SPR of *p*-polarization, (b) without deep-UV SPR of *s*-polarization and (c) conventional reflection image. Each image was captured with a laser power of 150 µW and exposure time of 10 sec. The spherical shape size of 5 µm was clearly observed using deep-UV SPR excitation. The autofluorescence intensities of yeast cells were successfully enhanced by deep-UV SPR. Referring to the conventional reflection image, several cells were not enhanced and observed in autofluorescence image. We considered that these cells floated from the substrate and were located out of range from the evanescent enhanced electric fields due to water immersion. Although autofluorescence is a very weak signal, and it is difficult to detect in the low-excitation laser power, high-sensitive images were obtained owing to the enhancement effect of deep-UV SPR.

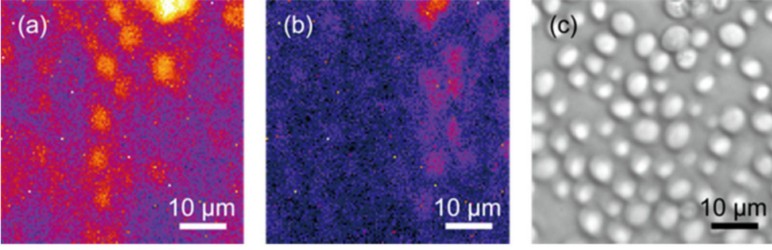

**Figure 5.** Water-immersed living yeast cells autofluorescence images on the aluminum thin film (**a**) with deep-UV SPR of *p*-polarization, (**b**) without deep-UV SPR of *s*-polarization and (**c**) conventional reflection image.

## 4. Discussion

The optimal aluminum thickness for deep-UV SPR excitation in the Kretschmann configuration is 21 nm [24]. The autofluorescence intensity was affected by the aluminum

thickness because SPR excitation efficiency and enhanced electric field intensity decrease as the aluminum thickness varies beyond the optimal range. The tolerance of aluminum thickness would be approximately 15–25 nm [24].

In Figure 4c observation, the tail shape of the fluorescent latexes is due to inadequate aperture adjustment of white-light illumination. Several parts of the different cells were observed between Figure 5a,c, which are considered to be caused by cell floating movement during the observation period. The different intensities in Figure 5a,b are also expected to be caused by cell floating. In the Kretschmann configuration, the electric field intensities decay exponentially with distance from the metal surface. High fluorescence intensity is observed when cells are nearby the surface. On the other hand, when cells are far away from the surface, the intensity becomes low.

## 5. Conclusions

In conclusion, theoretical and experimental investigations into deep-UV SPR excitation on an aluminum thin film through a sapphire prism were conducted. Numerical calculations were first performed to optimize the aluminum thickness and the incident angle of the light. By comparing the quartz and sapphire prism, the prism of sapphire shows a great advantage for high-sensitive bio-imaging of deep-UV SPR excitation with water-immersed specimens.

Fluorescence images of water-immersed polystyrene latexes excited with *p*-polarization and *s*-polarization were demonstrated. Individual latexes were clearly observed, and the fluorescence intensity was significantly higher than without deep-UV SPR. A 2.8-fold increase was confirmed when the fluorescence intensities were compared to the peak intensity. It was shown that deep-UV SPR enhances the fluorescence of water-immersed samples and enables highly sensitive observation.

Experimental measurements demonstrated the autofluorescence of living yeast cells, which were observed using deep-UV SPR. Moreover, deep-UV SPR led to a significant increase in autofluorescence intensity. Deep-UV SPR enhanced the autofluorescence of the cell structures, and the yeast cells were observed to exhibit high sensitivity. Thus, it can be concluded that a sapphire-prism-based Kretschmann configuration excites SPR in deep-UV in water-immersed specimens. On the basis of these results, deep-UV SPR can be considered an effective technique for achieving high-sensitivity observation of biological samples.

**Author Contributions:** Conceptualization, A.O. and Y.K.; methodology, A.O., W.I. and Y.K.; calculation, C.N.H.C.L. and H.M.; validation, C.N.H.C.L., H.M. and K.K.; data curation, C.N.H.C.L. and H.M.; writing original draft preparation, C.N.H.C.L.; funding acquisition, Y.K. All authors have read and agreed to the published version of the manuscript.

**Funding:** Core Research for Evolutional Science and Technology, Japan Science and Technology Agency (JPMJCR2003).

**Institutional Review Board Statement:** Not applicable.

**Informed Consent Statement:** Not applicable.

**Data Availability Statement:** Not applicable.

**Conflicts of Interest:** The authors declare no conflict of interest.

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
