# Peer review of "Autofluorescence Imaging of Living Yeast Cells with Deep-Ultraviolet Surface Plasmon Resonance"

_photonics, doi:10.3390/photonics9060424_

Round 1

Reviewer 1 Report

Referee Report on the paper “Autofluorescence Imaging of Living Yeast Cells with deep UV SPR” by С. N. H. С. Lah et al. submitted to Photonics.

In the paper, the authors report imaging of living yeast cells in water when irradiated by 266 nm light in the conditions of SPR excitation. 21 nm- thick aluminium films were used, the presence of a few nm-thick layers of Al oxide onto them is duly taken into account. This work may be considered as a direct continuation of the paper published in Anal. Chem. by (partly) the same authors in 2016, but it contains enough new material to be published.

In my opinion, this is a good and well-written paper. I recommend acceptance.   The only one drawback I noticed is the strange writing of the word “eukaryotic” on line 133. This is easy to remove.

Author Response

Thank you very much for your acceptance decision. We are very sorry for our mistype of eukaryotic”, and we removed this word.

Reviewer 2 Report

comment in attachment.

Author Response

Thank you very much for your valuable comments. Please see the attachment.

Reviewer 3 Report

The manuscript experimentally presented the deep UV-SPR with the aluminium coated sapphire prism for autofluorescence imaging of yeast cells. The expreimental work is clear to show the feasibility of the autofluorescence imaging with deep UV-SPR. I have a small concern about the structural parameter. The manuscript showed the results of 21nm thick aluminium film. Different aluminium film thickness will cause different resonance angle. Then, does the aluminium film thickness affect the autofluorescence intensity?

Author Response

(The authors gave the same response as above.)

Reviewer 4 Report

In the manuscript entitled “Autofluorescence Imaging of Living Yeast Cells with Deep Ul- 1traviolet Surface Plasmon Resonance”, the authors proposed a novel method for imaging free-labeled living yeast cells. Sapphire with a high refractive index prism was used to excite deep-UV SPR in the Kretschmann configuration. However, the experimental part of this manuscript is too simple, and there is no detailed analysis of the pictures. The manuscript can be further decided if the authors can address these concerns and make some improvements for details.

1.      In Figure 4c, the conventional reflection image shows the majority of fluorescent polystyrene latexes have tails. It looks markedly different from Figure 5c.

2.      In Figure 5, some cells are observed in the p-polarization deep-UV SPR image, and the others are observed in the s-polarization deep-UV SPR image. Can the author give an explanation? At the same time, the author should also explain the different intensities in Figures 5a and 5b.

3.      The authors should provide comparative cell imaging results of quartz prism and sapphire prism.

4.      Will the authors apply this finding to further studies beyond the detection of spontaneous fluorescence in cells?

Author Response

(The authors gave the same response as above.)

Round 2

Reviewer 4 Report

Critical Review on Photonics (photonics-1748347)

Decision: Accept

General comment:

In the manuscript entitled “Autofluorescence Imaging of Living Yeast Cells with Deep Ul- 1traviolet Surface Plasmon Resonance”, the authors proposed a novel method for imaging free-labeled living yeast cells. The authors have put considerable effort into addressing the reports of the referees. As a result, the paper is very much improved and I have no problem recommending it for publication.
